# Effect of Organic Amendments and Nano-Zinc Foliar Application on Alleviation of Water Stress in Some Soil Properties and Water Productivity of Barley Yield

Tamer H. Khalifa [1,*], Samah A. Mariey [2], Zeinab E. Ghareeb [3], Ismael A. Khatab [4] and Amal Alyamani [5]

1 Soil Improvement and Conservation Research Department, Soil, Water and Environment Research Institute (SWERI), Agriculture Research Center (ARC), Giza 12112, Egypt

2 Barley Research Department, Field Crops Research Institute, Agriculture Research Center (ARC), Giza 12619, Egypt; samahmaerie@gmail.com

3 Central Laboratory for Design and Statistical, Analysis Research, Agriculture Research Center (ARC), Giza 12622, Egypt; zeinabghareeb@yahoo.com

4 Department of Genetics, Faculty of Agriculture, Kafr El-sheikh University, Kafr El-sheikh 33516, Egypt; ismail.khatab@agr.kfs.edu.eg

5 Department of Biotechnology, Faculty of Science, Taif University, Taif 888, Saudi Arabia; a.yamani@tu.edu.sa

* Correspondence: tamer.khalifa@arc.sci.eg; Tel.: (+2)01060822017

**Abstract:** The scarcity of water resources in arid and semi-arid areas is considered a threat to agricultural sustainability. Therefore, approaches are needed to rationalize use of irrigation water without reducing crop productivity or degrading soil properties. The objective of this study was to investigate the effect of different organic amendments ($O_1$ = control, $O_2$ = compost, and $O_3$ = vermicompost) combined with different rates of nano- zinc foliar spraying ($Zn_1$ = 0, $Zn_2$ = 1 and $Zn_3$ = 2 gm/L), under irrigation supplements ($I_1$ = 100%, $I_2$ = 85%, and $I_3$ = 65% of water requirements) on clay soil characteristics, on the production of Egyptian barley Giza 126. Over two successive winter growing seasons, 2018/2019 and 2019/2020, field experiments were conducted as a split-split plot design with three replications. The results show that using vermicompost is an appropriate organic amendment to improve the physical and chemical properties of soils as compared with compost. Application of vermicompost led to a reduction in soil salinity (ECe), exchangeable sodium percentage (ESP), and soil bulk density (BD), of −5.67%, −5.44%, and −2.21%, respectively; there was a significant increase in soil organic carbon (SOC), available nitrogen (A.N), and field capacity (F.C.), of 43.75%, 14.37%, and 18.65%, respectively, compared with unamended soil ($O_1$). The maximum values for grain yield were increased by 13.2% and 14.9% in both seasons, respectively, and the irrigation water productivity of barley was increased more than compost and control. Vermicompost increased the irrigation water productivity for grain (1.69 and 1.69 kg grain $m^{-3}$) and straw (1.23 and 1.17 kg straw $m^{-3}$) in the first and second season, respectively. Similar trends were also observed from treatments on the water applied, stored water, and water application efficiency. Application of vermicompost and nano-Zn foliar spraying could be exploited for the development of barley growth and yield, which are enhanced under water-saving irrigation strategies.

**Keywords:** barley yield; compost; irrigation supplements; nano-zinc foliar spraying; vermicompost; water productivity

## 1. Introduction

Water availability in semiarid regions is endangered, not only due to changing climate conditions but also to human activities and land-use changes [1]. At a regional scale, Egypt is considered poor in water resources, while the flow of Nile water to Egypt will be reduced by as much as 25% during the period of filling the reservoir upstream of the dam [2]. In addition, Egypt has long dry summers and relatively short winters (December–February).

Moreover, there has been an upsurge of the population in the past few decades at a rate of increase near 2.5%. This, along with the rapid growth of human activities, has caused substantial changes in the environment, sometimes in damaging ways [3]. Drought in arid and semiarid regions is the most devastating environmental constraint causing much more yield loss than any other abiotic stress [4]. It occurs in virtually all climate regions, with drought-prone areas accounting for 16.2–41.2% of the world's arable land [5]. It adversely affects soil properties, plant growth, and overall productivity [6]. Drought prevents growth by decreasing the volume of water in plant cells, which interferes with the biochemical and physiological processes of plants [7].

The best approach is to use species that are tolerant of drought, through suitable varietal choice. Barley (*Hordeum vulgare* L.) is among the crops considered as the most drought tolerant of the small grain cereals and is a major crop in Mediterranean countries [8]. It is of high nutritional value, as it can be used with wheat in the bread industry. Globally, barley occupies the fourth rank in the cereal crops after wheat, rice, and maize [9,10]. Egypt also ranks first in the Arab world in terms of the area planted with barley, with 1187.2 thousand hectares, and the productivity is about 244 kg ha$^{-1}$ [9]. Although Egypt's barley production has fluctuated considerably over the last few years, it has tended to increase over the period 1971–2020, reaching 108,000 tons in 2020 [11].

Previous studies have indicated that the application of organic amended soils (with compost and vermicompost) helps crops to overcome the negative effects of drought [12,13]. Applying compost as organic amendment improved some soil physical and hydro-physical properties, with increased total soil porosity, void ratio, and soil moisture content (saturation percent, water field capacity, wilting point, available water) and decreased soil bulk density and water consumption [14–17]. Vermicomposting is a bio-oxidative process that uses earthworms and microorganisms for biochemical degradation, creating compost rich in humus, macronutrients, and micronutrients. It can improve the soil health status, enhance crop production, and improve the physical properties of the soil [18,19]. It has highly porous, allows high ventilation, and water storage capacity also, enriching the soil macro and micronutrients. This can lead to greater nutrient uptake and improved drought tolerance in comparison with conventional compost due to its humus content [20,21]. Application of vermicompost as a soil amendment improved soil properties and soil fertility by increasing soil organic matter, cation exchange capacity, and nutrient contents [22–25].

Zinc (Zn) plays an important role in the biosynthesis of different plant growth hormones such as auxins [26]. Nano fertilizers or nano-encapsulated nutrients may have properties that are efficient for crops releasing the nutrients on-demand; they control the release of chemical fertilizers that regulate plant growth and enhance the target activity [27]. Nanoparticles (nano-scale particles or NSPs) are molecular accumulations of 1 to 100 nm in one or more dimensions [28]. Nano-fertilizers have been designed to offer an effective new alternative to conventional fertilizers. The characteristics of the nanoparticles (increased surface area) allow these nanoparticles to increase their reactive points, causing changes in the absorption of these fertilizers into plants [29,30].

The purpose of this study was to evaluate the application of organic amendments, and nano- zinc, as well as their interactions, on alleviation of water stress impact on some soil properties and on water productivity in terms of barley yield.

## 2. Materials and Methods

### 2.1. Materials

Grains of Egyptian barley (*Hordeum vulgare* L.) Giza 126 cultivars were provided by Barley Res, Field Crops Res., Institute, Agricultural Research Centre in Egypt. Giza 126 properties include six rows, hulled accession, late heading, high height, moderated yield ability, precociousness, moderate productivity in favorable conditions, and tolerance to drought and fungal diseases. It originated from the cross of (Baladi Bahteem/S D729-Por12762-BC), ARC- Egypt released in 1995 [31,32].

## 2.2. Field Experiment Site

A field experiment was carried out on clay soil at Sakha Agric. Res. Station Farm, Kafr El-Sheikh Gov., in the North Nile Delta of Egypt, located at 31°05′36.28″ N, 30°56′53.56″ E (with an elevation 6 m above mean sea level) during two consecutive winter growth seasons, namely, 2018/2019 and 2019/2020.

Monthly meteorological data of air temperature (T, °C), relative humidity (RH, %), wind speed (Ws, m/s at 2 m height), and rainfall (mm month$^{-1}$) for both growing seasons were recorded at the weather station belonging to the Sakha Agrometeorological Station, Kafr EL-Sheikh Gov., Egypt (Figure 1).

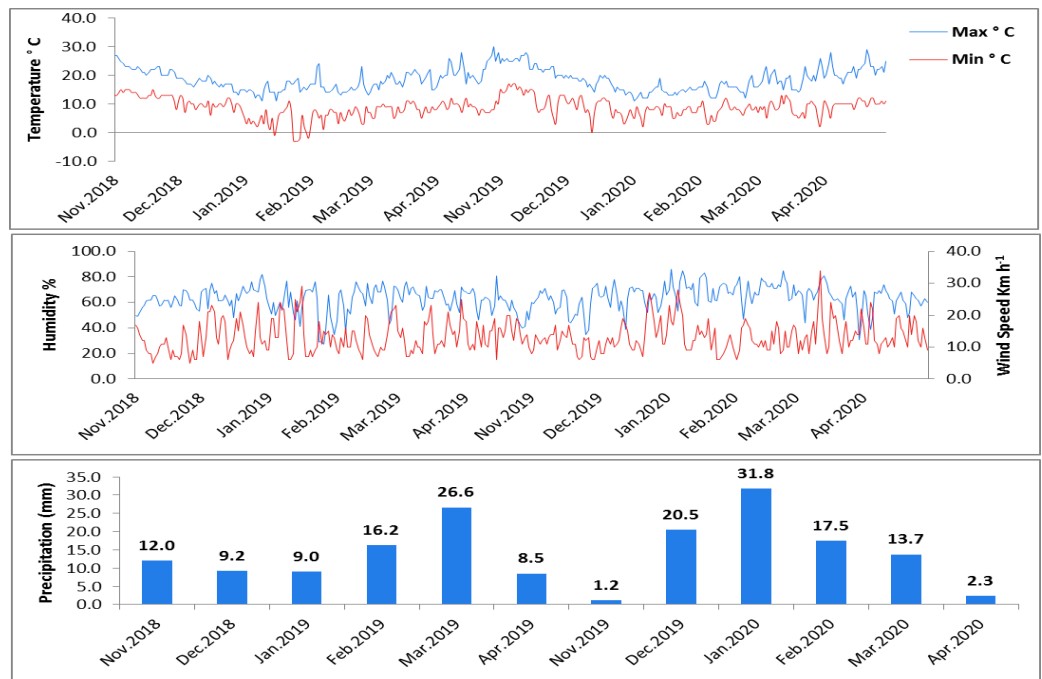

**Figure 1.** Monthly meteorological during the two winter growing seasons.

Soil samples were obtained from the surface layer (30 cm depth) using an auger for analysis of soil properties, before sowing. The initial physicochemical properties are presented in Table 1.

**Table 1.** Soil physical, chemical, and moisture properties before sowing.

| Physical Soil Properties | | | | | | | | |
|---|---|---|---|---|---|---|---|---|
| Soil Moisture Characteristics (%) | | | BD (Mg m$^{-3}$) | T.P (%) | Particle Size Distribution (%) | | | |
| F.C. | W.P. | A.W. | | | Sand | Silt | Clay | Texture |
| 41.51 | 21.59 | 20.53 | 1.35 | 49.06 | 17.96 | 25.13 | 56.92 | Clayey |
| Chemical Soil Characteristics | | | | | | | | |
| pH | ECe (dS m$^{-1}$) | ESP (%) | CEC (cmol kg$^{-1}$) | OM (%) | CaCO$_3$ (g kg$^{-1}$) | N | P | K |
| | | | | | | (mg kg$^{-1}$) | | |
| 8.38 | 4.81 | 10.15 | 37.93 | 0.86 | 2.61 | 30.89 | 8.13 | 255.13 |

F.C.: Field Capacity; W.P.: Wilting Point; A.W.: Available Water; BD: Bulk Density; T.P: Total porosity.

## 2.3. Field Experiment Design

The experimental design was a split-split plot design with three replications. The main plots included three levels of irrigation supplements (100%, 85%, and 65% of water requirements); subplots consisted of three organic amendments (without addition, compost

at a rate of 10 Mg ha$^{-1}$, and vermicompost at a rate of 5 Mg ha$^{-1}$). The sub-sub plots were occupied by three nano-zinc foliar spraying doses, (without, 1 and 2 gm nano Zn/L water). Nano-zinc foliar spraying was added, in two doses: after 45 and after 60 days of sowing.

Compost was made from rice straw and obtained from the Microbiology Res. Dep., at Sakha Agric. Res. Station, Kafr EL-Sheikh Gov., Egypt. Vermicompost was obtained from Tanta Uni., El-Gharbia Gov., Egypt. It was made from rice straw and animal wastes, with earthworm species *Eiseniafetida* and *DendrobaenaVeneta*. The chemical composition of compost and vermicompost is listed in Table 2.

**Table 2.** Chemical compositions of compost and vermicompost.

| Raw Materials | pH * | ECe * (dS m$^{-1}$) | O.M (%) | O.C (%) | C/N Ratio | Total Nutrients (%) | | |
|---|---|---|---|---|---|---|---|---|
| | | | | | | N | P | K |
| Compost | 7.71 | 4.09 | 26.89 | 15.60 | 18.00 | 1.75 | 0.92 | 1.25 |
| Vermicompost | 7.62 | 4.59 | 31.92 | 18.56 | 11.46 | 1.69 | 1.26 | 1.31 |

* pH and ECe were determined in soil, water suspension (1:10).

Nano-ZnO particles were prepared using the zinc acetate precursor method via sol-gel technique at Soil Dep., Agric. Fac., Tanta Uni., El-Gharbia Gov., Egypt, according to [33]. The structural phase of Nano-ZnO was characterized by an X-ray diffractometer (XRD), while the size and particle morphology were visualized via the transmission electron microscope (TEM) image technique. Figure 2 illustrates the crystalline size and purity of synthesized Nano-ZnO. The peaks show the prepared N-ZnO to be within the nano range. Laboratory prepared N-ZnO particles are not exactly circular with size ranging from 30 to 35 nm of the hexagonal quartzite structures.

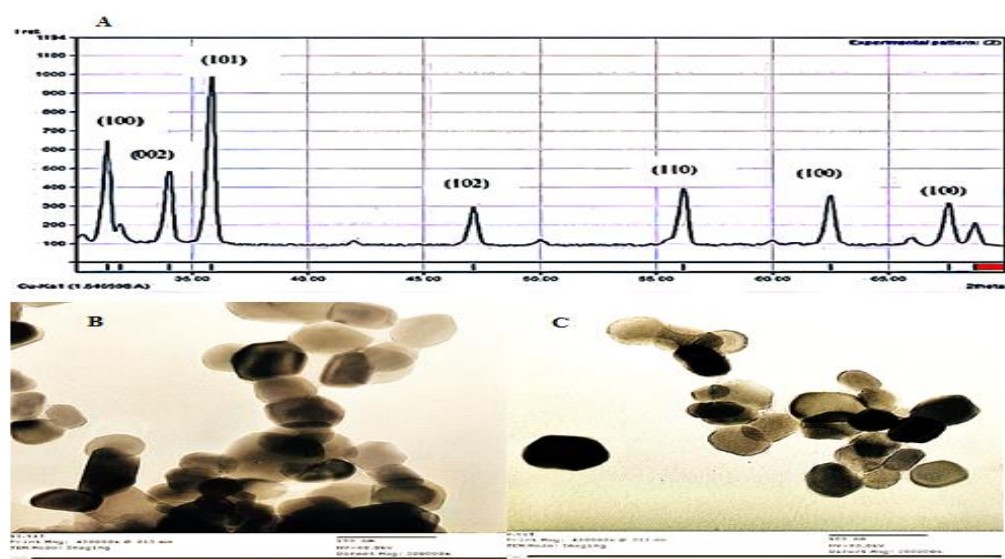

**Figure 2.** (**A**) The X-ray diffractometer pattern of nano-ZnO powder. (**B**) The TEM images display the diffusion and structural characteristics. (**C**) Shows the laboratory prepared Nano-ZnO particles.

The area of each plot was 3 m × 4 m (12 m$^2$). The soil organic amendments (compost and vermicompost) and calcium superphosphate (15.5% P$_2$O$_5$) at the rate of 238 kg ha$^{-1}$ were added to the tillage process, directly before sowing. Barley grains were sown at the rate of 119 kg ha$^{-1}$ using a broadcasting method on 19 November in both seasons. The normal cultural practices for growing barley were applied according to the Ministry of Agriculture's recommendations. Description of the factorial treatments and their codes are shown in Table 3.

**Table 3.** Description of the used factorial treatments and their codes.

| No | Irrigation Supplements | Organic Amendments | Nano-Zinc Foliar Spraying | Symbol | Bi-Plot |
|---|---|---|---|---|---|
| 1 | | Control (O$_1$) | Without (Zn$_1$) | I$_1$O$_1$ZN$_1$ | T$_1$ |
| 2 | | | 1 g/L (Zn$_2$) | I$_1$O$_1$ZN$_2$ | T$_2$ |
| 3 | 100% Full water requirement (I$_1$) | | 2 g/L (Zn$_3$) | I$_1$O$_1$ZN$_3$ | T$_3$ |
| 4 | | Compost (O$_2$) | Without (Zn$_1$) | I$_1$O$_2$ZN$_1$ | T$_4$ |
| 5 | | | 1 g/L (Zn$_2$) | I$_1$O$_2$ZN$_2$ | T$_5$ |
| 6 | | | 2 g/L (Zn$_3$) | I$_1$O$_2$ZN$_3$ | T$_6$ |
| 7 | | Vermicompost (O$_3$) | Without (Zn$_1$) | I$_1$O$_3$ZN$_1$ | T$_7$ |
| 8 | | | 1 g/L (Zn$_2$) | I$_1$O$_3$ZN$_2$ | T$_8$ |
| 9 | | | 2 g/L (Zn$_3$) | I$_1$O$_3$ZN$_3$ | T$_9$ |
| 10 | | Control (O$_1$) | Without (Zn$_1$) | I$_2$O$_1$ZN$_1$ | T$_{10}$ |
| 11 | | | 1 g/L (Zn$_2$) | I$_2$O$_1$ZN$_2$ | T$_{11}$ |
| 12 | 85% of water requirement (I$_2$) | | 2 g/L (Zn$_3$) | I$_2$O$_1$ZN$_3$ | T$_{12}$ |
| 13 | | Compost (O$_2$) | Without (Zn$_1$) | I$_2$O$_2$ZN$_1$ | T$_{13}$ |
| 14 | | | 1 g/L (Zn$_2$) | I$_2$O$_2$ZN$_2$ | T$_{14}$ |
| 15 | | | 2 g/L (Zn$_3$) | I$_2$O$_2$ZN$_3$ | T$_{15}$ |
| 16 | | Vermicompost (O$_3$) | Without (Zn$_1$) | I$_2$O$_3$ZN$_1$ | T$_{16}$ |
| 17 | | | 1 g/L (Zn$_2$) | I$_2$O$_3$ZN$_2$ | T$_{17}$ |
| 18 | | | 2 g/L (Zn$_3$) | I$_2$O$_3$ZN$_3$ | T$_{18}$ |
| 19 | | Control (O$_1$) | Without (Zn$_1$) | I$_3$O$_1$ZN$_1$ | T$_{19}$ |
| 20 | | | 1 g/L (Zn$_2$) | I$_3$O$_1$ZN$_2$ | T$_{20}$ |
| 21 | 65% of water requirement (I$_3$) | | 2 g/L (Zn$_3$) | I$_3$O$_1$ZN$_3$ | T$_{21}$ |
| 22 | | Compost (O$_2$) | Without (Zn$_1$) | I$_3$O$_2$ZN$_1$ | T$_{22}$ |
| 23 | | | 1 g/L (Zn$_2$) | I$_3$O$_2$ZN$_2$ | T$_{23}$ |
| 24 | | | 2 g/L (Zn$_3$) | I$_3$O$_2$ZN$_3$ | T$_{24}$ |
| 25 | | Vermicompost (O$_3$) | Without (Zn$_1$) | I$_3$O$_3$ZN$_1$ | T$_{25}$ |
| 26 | | | 1 g/L (Zn$_2$) | I$_3$O$_3$ZN$_2$ | T$_{26}$ |
| 27 | | | 2 g/L (Zn$_3$) | I$_3$O$_3$ZN$_3$ | T$_{27}$ |

*2.4. Studied Traits*

2.4.1. Agronomical Parameters

At harvest time six traits were measured. Plant height (PH) was measured on a random sample of five plants in each plot as the length from the soil surface to the tip of the spike. Thousand kernel weights (TKWs) were calculated by weighing 1000 grains randomly from each treatment. The grain yield (GY) was determined by harvesting the yield of the central area (1.6 m$^2$) of the plot, and then transformed to the unit of (t ha$^{-1}$). Biological yield was determined as the total biomass of the harvested plants (kg plot$^{-1}$), then it was transformed into (t ha$^{-1}$). After threshing, the yield of straw was calculated (biological - grain yield) for each plot (t ha$^{-1}$). Harvest index (HI) calculation in this investigation was as follows:

$$\text{H.I.} = \frac{\text{Grain Yield (t/ha)}}{\text{Biological yield (t/ha)}} \times 100 \tag{1}$$

2.4.2. Water Relations Parameters

The amount of irrigation water applied to barley as a winter crop in each irrigation was determined based on reaching the soil moisture content to its field capacity multiplied by the irrigation efficiency (55%). At the time of irrigation, soil samples were collected regularly using an auger at successive depths of 15 cm to 60 cm until it achieved the permissible moisture level (50% of available soil moisture).

Water applied (WA) was equal to irrigation water plus total rainfall.

Stored water (WS) was calculated according to [34] using the following equation:

$$WS = \sum_{i=1}^{i=n} \frac{Q2 - Q1}{100} \times D \times BD \tag{2}$$

where:

$WS$ = Stored water, cm, and transfer to $m^3\ ha^{-1}$

$Q_2$ = Soil layer moisture content, wt/wt%, 48 h after irrigation.

$Q_1$ = Soil layer moisture content, wt/wt%, just before the same irrigation.

$D$ = Effective root zone, 60 cm.

Bed = Bulk density of the soil layer, Mg $m^{-3}$

Water application efficiency (Ea): it is the ratio between the amount of stored water ($m^3\ ha^{-1}$) and water applied ($m^3\ ha^{-1}$) as described by [35]:

$$Ea = (WS/WA) \times 100 \tag{3}$$

Irrigation water productivity (PIW, kg $m^{-3}$): it is the ratio between the grain and straw yields (kg $ha^{-1}$) and water applied ($m^3\ ha^{-1}$), calculated according to [36]:

$$PIW\ (kg\ m^{-3}) = (Y/WA) \times 100 \tag{4}$$

2.4.3. Soil Properties

After harvest, soil samples were obtained from the surface layer (30 cm depth) using an auger and analysis of some soil properties. The soil samples were dried, ground, and passed through a 2-mm sieve. Soil reaction (pH) was estimated in a 1:2.5 suspension (w:v for soil:distilled water) with a pH meter (Genway, UK). Soil electrical conductivity (ECe dS $m^{-1}$) was identified in the soil paste extract, with an EC meter (Genway, UK). The exchangeable sodium percentage (ESP) was measured as described by [37,38]. Cation exchange capacity (CEC, cmol $kg^{-1}$) was determined using the ammonium acetate method. as described by [39]. Prganic matter (O.M, %) and soil organic carbon (SOC, %) content were determined according to [40]. Total $Ca^{+2}$ carbonate ($CaCO_3$, g $kg^{-1}$) was determined volumetrically using a Collins calcimeter [41]. The soil content of available N was determined using $K_2SO_4$ (1%) according to [42]. Available P and K were extracted by ammonium bicarbonate- DTPA and determined according to [43]. Soil physical traits and moisture were determined in undisturbed soil samples following previously explained methods [44–46].

*2.5. Data Analysis*

Analysis of variance (ANOVA), appropriate for the split-split plot design, was performed using MSTATC statistical package (MSTAT-C with MGRAPH version 2.10, Crop and Soil Sciences Department, Michigan State University, East Lansing, MI, USA) [47]. Differences among means were tested by the least significant difference (LSD) test at a 5% probability level. GGE-biplot (genotype main effect plus genotype-by-environment interaction) uses the first two principal components (PC1 and PC2) to display the two-way data in the biplot graph [48]. The biplot presented in this paper was performed using the procedures of the GenStat ver. 12 software, Oxford, UK, 2008. [49]. Pearson's correlation analysis was performed using SPSS 22.0 version (SPPS Inc., Chicago, IL, USA).

**3. Results and Discussion**

*3.1. Effect of Organic Soil Amendments and Nano-Zn Foliar Spraying on Agronomical Traits under Irrigation Supplements*

This study investigated significant differences among the studied organic amendments and nano-Zn foliar spraying under different irrigation supplement treatments, in relation to biological yield (BY, t $ha^{-1}$), grain yield (GY, t $ha^{-1}$), straw yield (SY t $ha^{-1}$), 1000-kernels weight (TKW g), plant height (PH cm) and harvest index (HI, %) in 2018/19 and 2019/20 (Table 4).

**Table 4.** Effect of different organic amendments and nano-Zn foliar spraying concentrations under irrigation supplements on agronomical traits in two growing seasons 2018/19 and 2019/20.

| Season | 1st Season, 2018/2019 | | | | | | 2nd Season 2019/2020 | | | | | |
|---|---|---|---|---|---|---|---|---|---|---|---|---|
| Trait Treatment | $BY_1$ | $GY_1$ | $SY_1$ | $TKW_1$ | $PH_1$ | $HI_1$ | $BY_2$ | $GY_2$ | $SY_2$ | $TKW_2$ | $PH_2$ | $HI_2$ |
| Irrigation supplements (I) | | | | | | | | | | | | |
| $I_1$ | 8.80 | 5.04 | 3.76 | 58.07 | 93.85 | 57.25 | 8.94 | 5.23 | 3.71 | 59.54 | 96.22 | 58.48 |
| $I_2$ | 8.61 | 4.89 | 3.72 | 57.93 | 91.81 | 56.76 | 8.74 | 5.10 | 3.64 | 59.38 | 94.22 | 58.31 |
| $I_3$ | 8.28 | 4.75 | 3.53 | 57.78 | 89.81 | 57.31 | 8.42 | 4.95 | 3.47 | 59.25 | 91.89 | 58.72 |
| F-test | ** | * | NS | ** | ** | NS | ** | * | NS | ** | ** | NS |
| LSD | 0.091 | 0.189 | - | 0.029 | 1.275 | - | 0.101 | 0.142 | - | 0.017 | 1.023 | - |
| Organic amendments (O) | | | | | | | | | | | | |
| $O_1$ | 8.47 | 4.63 | 3.65 | 55.84 | 89.26 | 55.90 | 8.42 | 4.84 | 3.58 | 57.72 | 92.33 | 57.47 |
| $O_2$ | 8.91 | 4.95 | 3.64 | 58.93 | 92.22 | 57.59 | 8.72 | 5.14 | 3.58 | 60.20 | 94.11 | 58.93 |
| $O_3$ | 9.02 | 5.11 | 3.72 | 59.01 | 94.00 | 57.83 | 8.96 | 5.30 | 3.66 | 60.25 | 95.89 | 59.10 |
| F-test | ** | ** | NS | ** | ** | ** | ** | ** | NS | ** | ** | * |
| LSD | 0.026 | 0.099 | - | 1.262 | 1.666 | 1.153 | 0.026 | 0.097 | - | 1.277 | 1.896 | 1.101 |
| Nano-Zn foliar spraying (Zn) | | | | | | | | | | | | |
| $Zn_1$ | 8.56 | 4.52 | 3.82 | 57.14 | 86.81 | 54.14 | 8.48 | 4.69 | 3.79 | 58.80 | 89.22 | 55.33 |
| $Zn_2$ | 8.83 | 4.96 | 3.63 | 57.84 | 92.89 | 57.69 | 8.73 | 5.09 | 3.64 | 59.32 | 95.33 | 58.27 |
| $Zn_3$ | 9.01 | 5.21 | 3.55 | 58.80 | 95.78 | 59.49 | 8.90 | 5.51 | 3.39 | 60.05 | 97.78 | 61.91 |
| F-test | ** | ** | ** | ** | ** | ** | ** | ** | ** | ** | ** | ** |
| LSD | 0.036 | 0.094 | 0.12 | 0.521 | 0.871 | 1.25 | 0.041 | 0.102 | 0.117 | 0.592 | 1.216 | 1.237 |

Abbreviations: Biological yield (BY t ha$^{-1}$), grain yield (GY t ha$^{-1}$), straw yield (SY t ha$^{-1}$), 1000-kernels weight (TKW g), plant height (PH cm), and harvest index (HI%). *: refers to significant, and **: refers to highly significant differences between them ($p < 0.05, 0.01$).

Results presented in Table 4 show that the different irrigation supplement treatments (100% ($I_1$), 85% ($I_2$), and 65% ($I_3$)) significantly affected all observed traits except for straw yield and harvest index in both growing seasons. Decreasing the irrigation requirements from 100% ($I_1$) to 85% ($I_2$) and 65% ($I_3$) significantly reduced all studied traits and grain yield by (2.97%, 5.75%, 2.48% and 5.35%) in both growing seasons, respectively, as shown in Figure 3. These results were in agreement with [50,51], who confirmed that t deficit irrigation significantly reduced the yield traits compared with unstressed ones.

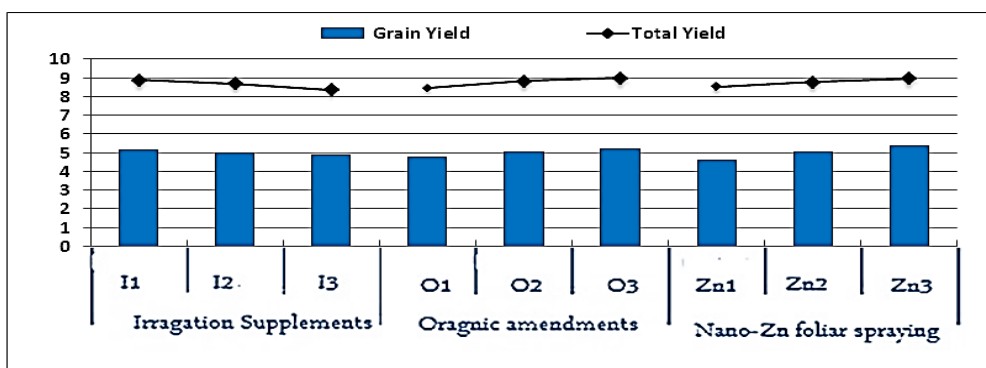

**Figure 3.** Effect of different organic amendments and nano-Zn foliar spraying concentrations under irrigation supplements on some barley grain and biological yield across the two growing seasons 2018/19 and 2019/20.

Table 4 illustrated that there were significant differences among studies of organic amendment treatments (control ($O_1$), compost ($O_2$), and vermicompost ($O_3$)) for all observed traits except for straw yield in both seasons, with the vermicompost application being more effective than compost. Application of vermicompost led to a significant increased GY (13.2% and 14.9%) in both seasons, respectively, compared to control as shown in

Figure 3. Maximum values for all barley yield traits compared to other treatments were recorded with the vermicompost treatment: values for BY, GY, SY, TKW, PH and HI in both seasons were 9.02–8.96 t ha$^{-1}$; 5.11–5.30 t ha$^{-1}$; 3.72–3.66 t ha$^{-1}$; 59.01–60.25 g; 94.00–95.89 cm and 57.83–59.10%, respectively. However, the differences between compost (O$_2$) and vermicompost (O$_3$) were not great enough to score the significance level for the 1000-kernels weight (85.93 and 59.01 g −60.20 and 60.25 g) and harvest index (57.59 and 57.83–58.93 and 59.10%) in both seasons, respectively. Therefore, compost treatment (O$_2$) could be used to donate the approximate effect (with no significant difference) of vermicompost (O$_3$) for the TKW and HI. Similar differences in organic treatments were recorded by [52–54] which found a positive effect of organic amendments on crop yields; the authors attribute this t to a critical role for organic matter in the soil ecosystem. Soil organic matter plays a dynamic part in improving soil fertility and quality by increasing the soil's capacity to accumulate and supply vital nutrients; it improves soil structure through improved water holding capacity and can improve the activity of microorganisms in the soil and enhance biodiversity [55].

Regarding the effect of Nano-Zn foliar spraying treatments, data in Table 4 revealed that there were significant differences among all studies in both seasons. Figure 3 illustrated that adding nano-zinc enhanced biological and grain yield compared to control. The best treatment of nano-zinc application was 2 g, which recorded the highest mean values of all traits except for straw yield in both seasons. Nano-zinc rates had a positive, high effect on BY (9.01–8.90 t ha$^{-1}$), GY (5.21–5.51 t ha$^{-1}$), TKW (58.80–60.05 g), PH (95.78–97.78 cm), and HI (59.49–61.91%) in both seasons, respectively. Meanwhile, increasing nano-zinc to 2 g produced the lowest straw yield with average 3.55–3.39 t ha$^{-1}$ 3.82–3.79 ton ha$^{-1}$ in both seasons, respectively compared to control. The obtained results of barley production and grain yield development could be explained by nano-Zinc fertilizers playing an important role in the regulation of plant growth and enhancement of plant bio-activities [27,56]. On the other hand, the straw yield reduction by adding nano-Zinc might be due to the decreased Zinc effect on biomass, demonstrating that the importance of nano-Zinc fertilizers for the barley grain set is greater than for vegetative growth. Similarly, results of improvement in different grain yield traits Zn supplements were obtained by [26].

The bilateral interaction effect of irrigation supplements, organic amendment, and nano-Zn foliar spraying on all agronomical traits is shown in Supplementary Table S1.

These results (Supplementary Table S1 and Figure 4) revealed that the interaction between irrigation supplements and organic amendment treatments had a highly significant effect only on biological yield traits in both seasons. The results showed that I$_1$O$_3$ treatment (adding vermicompost application under 100% supplement irrigation) gave the maximum values (9.02–9.16 t ha$^{-1}$, 5.20–5.38 t ha$^{-1}$, 3.82–3.78 t ha$^{-1}$, 59.15–60.39 g, and 96.00–98.00 cm) for BY, GY, SY, TKW and PH in both seasons, respectively. From the previously mentioned results, the vermicompost application increased the plant growth response for grain and biological yield under each irrigation treatment. These results are in agreement with those obtained by [57,58] who reported that using vermicompost had a positive effect on plant growth. The results indicated the importance of vermicompost as an organic amendment with water supplements effect on barley growth and yield development.

Concerning the interaction of irrigation supplements and nano-Zn foliar spraying, the results in Supplementary Table S1 show that there are no differences in significance for all traits in both seasons. However, I$_1$ Zn$_3$, I$_2$ Zn$_3$ and I$_3$ Zn$_3$ treatments (adding 2 g nano-Zn under different irrigation supplement treatments) recorded the best reading for most cases, pointing to benefits of increasing of nano-zinc doses under different irrigation supplements in terms of the barley yield traits. Furthermore, nano-zinc addition may affect the growth, development of barley plants. These results are in general agreement with those obtained by [59] who reported that the application of nano Zn-Fe oxide caused an 88% increase in the grain yield as compared to control under severe water limitation.

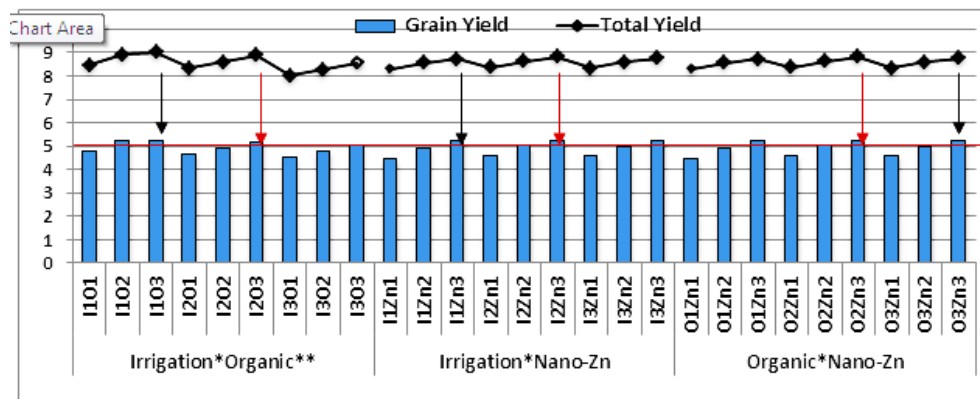

**Figure 4.** The bilateral interaction effect of irrigation supplements, organic amendment, and nano-Zn foliar spraying on yield (average data over the two studied seasons). *: refers to significant, and **: refers to highly significant differences between them ($p < 0.05$, 0.01).

For the interaction of organic amendment and nano-Zn foliar spraying rates, data shown in Supplementary Table S1 indicate highly significant differences for TKW and PH traits in both seasons. Meanwhile, the greatest values were obtained from $O_3 Zn_3$ treatment, containing vermicompost ($O_3$) with 2 g nano-Zn treatment ($Zn_3$) for all traits except for straw yield trait in both seasons. $O_3 Zn_3$ treatment recorded the maximum values 9.02–9.16 t ha$^{-1}$; 5.39–5.67 t ha$^{-1}$; 59.62–60.47 g; 97.33–98.78 cm, and 59.76–61.92% for BY, GY, TKW, PH, and HI in both seasons, respectively. The obtained results inducted that increasing both nano-Zn levels with vermicompost (across chemical compositions) enhance the barley yield traits with significantly increased biological yield, grain yield, 1000-kernels weight, and plant height.

The interaction among the three irrigation supplements, three organic amendments, and nano-Zn foliar spraying treatments as a total of 27 tri-treatments (Table 3) is shown in Supplementary Table S2 and graphically illustrated using a Treatment-Traits (TT) bi-plot in Figure 5.

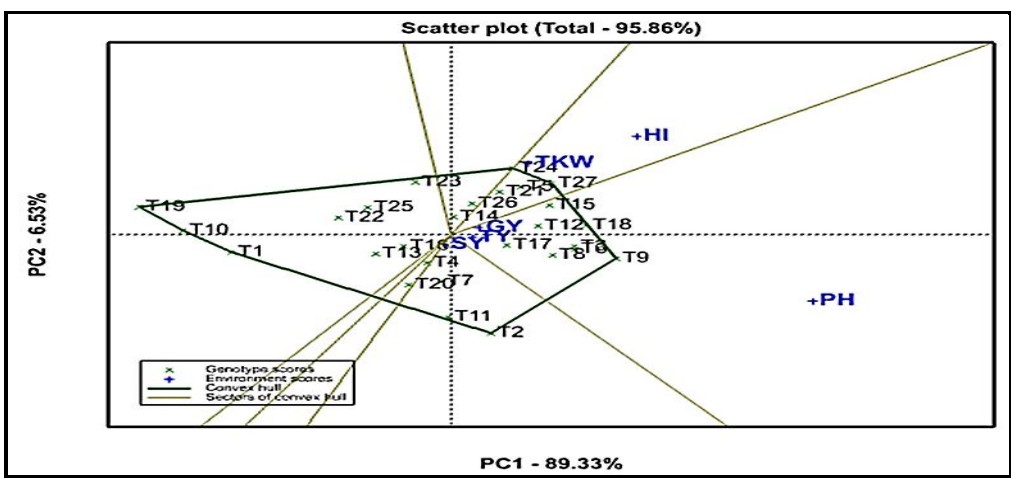

**Figure 5.** Polygon (which won where) view of the barley treatment-by-trait (TT) biplot of twenty-seven treatments for six traits. Abbreviations: biological yield (BY), grain yield (GY), straw yield (SY), 1000-kernels weight (TKW), plant height (PH), and harvest index (HI). $T_1$ to $T_{27}$ refer to Table 3.

The results in Supplementary Table S2 indicate that $I_1 O_3 Zn_3$ (adding vermicompost + 2 g nano-Zn under 100% irrigation requirements) recorded the greatest values for grain yield (5.52 t ha$^{-1}$) and 1000-kernel weight (59.76 g) in the 1st season and highest values for BY (9.24–9.38 t ha$^{-1}$) and tallest PH (99.33–100.67 cm) in both seasons, respectively. Moreover, GY had the highest value with $I_2 O_3 Zn_3$ (adding vermicompost + 2 g nano-Zn

under 85% irrigation supplements) with 5.73 t ha$^{-1}$ in the 2nd season. However, $I_1 O_3$ $Zn_2$ (adding vermicompost + 1 g nano-Zn under 100% irrigation supplements) gave the heaviest TKW (61.13 g) in the 2nd season. On the other hand, $I_1 O_1 Zn_1$ as a control (no adding an organic amendment or nano-Zn under 100% irrigation requirements) and $I_1 O_3$ $Zn_1$ (adding vermicompost + no nano-Zn under 100% irrigation supplements) produced the same effect on the straw yield, recording (3.94–3.90 ton ha$^{-1}$) over the two growing seasons, respectively. These results indicated that the decrease or absence of nano-Zn may be enhancing the straw growth in reverse to grain productivity, suggesting that nano-Zn is most importantly the final grain yield. In general, applying the organic amendment and foliar spraying of nano-Zn treatments gave the best barley growth and yield development compared to the control treatment ($I_1 O_1 Zn_1$) in this study. The results point to the role of vermicompost application in barley enzyme activities under drought stress, providing better conditions for the uptake of water and nutrients and enhancing grain filling [50,60].

It is noticeable that the Treatment-Traits (TT) bi-plot model using the average data over two seasons of the 27 different factorial treatments (Table 3) accounts for 95.86% of the total variation in the barley traits across treatments representing 89.33% and 6.53%, variance attributable to first two principal components for the PC1 and PC2 principal components, respectively (Figure 5). The results were in good harmony with [51–71] which reported that if both PC's reflected more than 60% of the total variation, then the TT bi-plot model achieved goodness of fit. The Treatment-Traits (TT) bi-plot model is used as a good tool to determine the effects of measured treatments on the multiple traits in the same bi-plot graph. In TT bi-plot, an ideal treatment has been defined as the treatment that combines several good traits in its composition.

Nine barley treatments generated a bi-plot ($T_2$, $T_9$, $T_{18}$, $T_{27}$, $T_{24}$, $T_{19}$, $T_{10}$, $T_1$, and $T_{11}$) were located on the right of the original points at the vertex of the polygon. Among the vertex treatments, $T_{18}$ and $T_9$ exhibited superior performance for the BY, GY, and PH, indicating that these treatments could be exploited for the development of barley yield that is distinguished in these traits. However, $T_{24}$ and $T_9$ revealed good behavior for the TKW and HI, that increasing nano-Zn may have a role in enhancing grain filling. Meanwhile, $T_2$ and $T_{11}$ were the best for straw yield (SY); adding combination treatments without organic matter may inhibit formation of grains as compared to straw. The other vertex fertilizer treatments $T_{19}$, $T_{10}$, and $T_1$ treatments located on the left side of the graph were not characterized for any trait. These treatments were inferior for all measured traits. These results indicate that the polygon view of the TT bi-plot is the best way to explain and summarize the interaction pattern between treatments and traits. The obtained results were similar to those obtained on barley by [62].

In Figure 6, treatments are ranked along the average tester axis (ATC) line that passes through the bi-plot origin and the average trait, with the arrow pointing to higher (small circle which is located on the line). The graph presents a vector view of the treatments-by-trait bi-plot representing the ranking of twenty-seven (irrigated-fertilizer combination) treatments based on their ideal mean performance over the several traits. In a TT bi-plot, $T_{18}$ ($I_2 O_3 Zn_3$, adding vermicompost + 2 g nano-Zn under 85% irrigation supplements) that was positioned closest to the center of the concentric circles was considered as the ideal treatment (best) based on measured traits. These concentric circles allow the comparison of all treatments with the ideal one, namely, $T_{18}$ ($I_2 O_3 Zn_3$). Therefore, the other near treatments were $T_9$, $T_3$, $T_6$, $T_{12}$, $T_{15}$, and $T_{8,}$ located in the direction of the increasing arrow. Therefore, it observed that the application of high amounts of nano-Zn was useful in obtaining desired micronutrient fertilizer application in barley production under different irrigation supplements.

On the other hand, $T_{19}$, $T_{10}$, $T_1$, $T_{22}$, and $T_{25}$ treatments are located below average with decreasing response. Treatments without nano-Zn foliar spraying ($zn_1$) recorded the lowest performance under different irrigation conditions for all traits. From the mentioned results, the polygon view as well as a vector view of the TT bi-plot technique gave an overall picture and summarized both the all measured treatments and trait knowledge, ranking

and determining the best treatments. The ideal treatment $T_{18}$ ($I_2 O_3 Zn_3$) thus recorded the best performance, indicating that adding vermicompost with 2 g nano-Zn can improve the barley growth and yield production in case of water deficiency to 85% irrigation requirements. Generally, the application of nano-zinc with vermicompost was useful in enhancing barley production under drought stress. These findings were in corroborated those of [56].

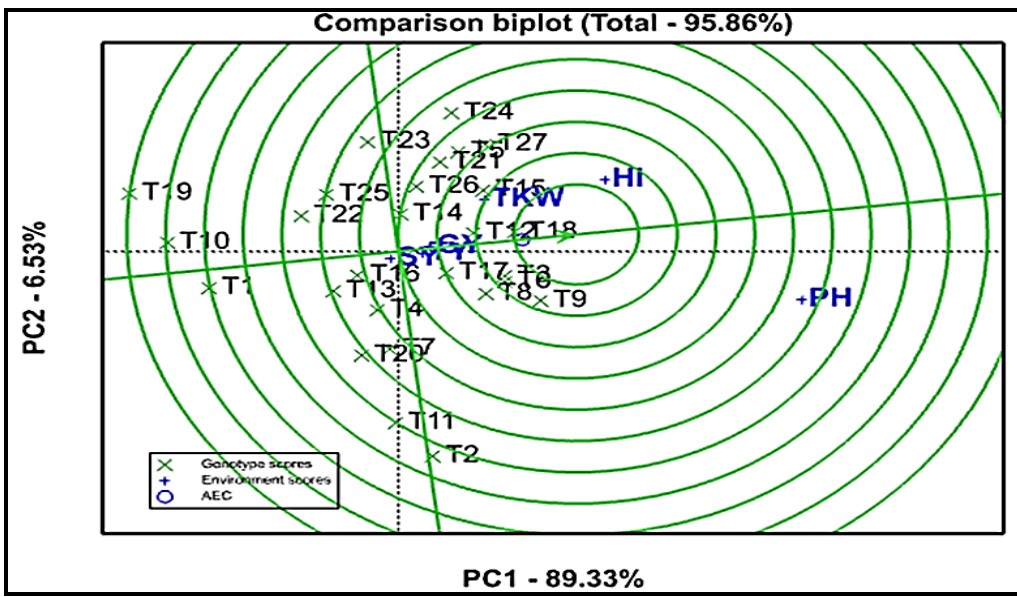

**Figure 6.** Ideal treatment of TT bi-plot, showing the ranking of twenty-seven barley irrigated-fertilizer combination treatments for various measured traits. Abbreviations: Biological yield (BY), grain yield ton ha$^{-1}$ (GY), straw yield (SY), 1000-kernels weight (TKW), plant height (PH), and harvest index% (HI). T1 to T 27 refer to Table 3.

### 3.2. Effect of Organic Soil Amendments and Nano-Zn Foliar Spraying on Water Relations Parameters

The effect of the irrigation supplements, organic amendment, and nano-Zn foliar spraying of the total water applied, stored water, water application efficiency, and irrigation water productivity for grain and straw is shown in Table 5 and Supplementary Figure S1. With irrigation supplement treatments $I_2$ and $I_3$ reduced the total water applied (500 and 936 m$^3$ ha$^{-1}$), (645 and 946 m$^3$ ha$^{-1}$) in the first and second seasons, respectively. Also, application of vermicompost reduced the total water applied (253 and 226 m$^3$ ha$^{-1}$) in the first and second seasons, respectively. Regarding nano-Zn foliar spraying, increasing the rate of foliar spraying increased the total water applied. Higher total water applied (4042 and 3909 m$^3$ ha$^{-1}$) was observed from full irrigation supplement ($I_3$) with nano-Zn foliar spraying at a rate of 2 g L$^{-1}$ ($zn_3$) in the first and second seasons, respectively. Simultaneously, similar trends were also observed from treatments on stored water and water application efficiency.

Irrigation water productivity of barley was affected by irrigation supplements, organic amendment, nano-Zn foliar spraying, and their interactions (Table 5). The higher irrigation water productivity for grain (1.64 and 1.79 kg grain m$^{-3}$) and straw (1.22 and 1.25 kg straw m$^{-3}$) in the first and second season, respectively, were observed with $I_3$. These results could be attributed to the significant differences between grain and straw yield of barley, and amounts of water supplied Similar results were obtained by [36,63]. Vermicompost enhances the irrigation water productivity of barley more than compost and control, the irrigation water productivity for grain (1.60 and 1.73 kg grain m$^{-3}$) and straw (1.16 and 1.20 kg straw m$^{-3}$) in the first and second season, respectively. This could be attributed to a decrease in the water applied due to saving soil moisture content. These findings are similar to those obtained by [21].

**Table 5.** The mean performance of total water applied, stored water, water application efficiency, and irrigation water productivity of grain and straw as affected by different treatments (irrigation supplements, organic amendments, and nano-Zn foliar spraying.

| Seasons | 1st, 2018/2019 | | | | | 2nd, 2019/2020 | | | | |
|---|---|---|---|---|---|---|---|---|---|---|
| Properties Treatment | WA | WS | Ea | PIW (Grain) | PIW (Straw) | WA | WS | Ea | PIW (Grain) | PIW (Straw) |
| Irrigation supplements (I) | | | | | | | | | | |
| $I_1$ | 3835 | 2825 | 73.64 | 1.32 | 0.98 | 3722 | 2877 | 77.29 | 1.41 | 1.00 |
| $I_2$ | 3335 | 2325 | 69.69 | 1.47 | 1.12 | 3077 | 2232 | 72.52 | 1.66 | 1.19 |
| $I_3$ | 2899 | 1889 | 65.11 | 1.64 | 1.22 | 2776 | 1931 | 69.53 | 1.79 | 1.25 |
| Organic amendments (O) | | | | | | | | | | |
| $O_1$ | 3491 | 2482 | 70.72 | 1.34 | 1.06 | 3326 | 2482 | 74.26 | 1.47 | 1.09 |
| $O_2$ | 3280 | 2270 | 68.79 | 1.49 | 1.13 | 3089 | 2244 | 72.26 | 1.65 | 1.18 |
| $O_3$ | 3238 | 2228 | 68.37 | 1.60 | 1.16 | 3100 | 2255 | 72.32 | 1.73 | 1.20 |
| Nano-Zn foliar spraying (Zn) | | | | | | | | | | |
| $Zn_1$ | 3298 | 2289 | 68.94 | 1.39 | 1.17 | 3136 | 2292 | 72.63 | 1.52 | 1.22 |
| $Zn_2$ | 3366 | 2356 | 69.59 | 1.49 | 1.09 | 3189 | 2344 | 73.10 | 1.62 | 1.16 |
| $Zn_3$ | 3405 | 2395 | 69.92 | 1.55 | 1.06 | 3249 | 2405 | 73.62 | 1.72 | 1.06 |

Abbreviations: W.A, Total water applied ($m^3\ ha^{-1}$); W.S, Stored water ($m^3\ ha^{-1}$); Ea, Water application efficiency (%) and PIW, Irrigation water productivity for grain and straw ($kg\ m^{-3}$). Rainfall in 2018/2019 = 815 $m^3\ ha^{-1}$ and 2019/2020 = 870 $m^3\ ha^{-1}$.

Nano-Zn foliar spraying at a rate of 2 g $L^{-1}$ ($zn_3$) increased the irrigation water productivity for grain (1.22 and 1.72 kg grain $m^{-3}$) in the first and second seasons, respectively, while the higher irrigation water productivity for straw (1.17 and 1.22 kg straw $m^{-3}$) were obtained without nano-Zn foliar spraying ($zn_1$) in the first and second seasons, respectively. This may be attributed to nano-zinc addition which improved the yield growth and production under water drought. These results are in general agreement with those obtained by [59].

The highest irrigation water productivity for grain (1.89 and 2.04 kg grain $m^{-3}$) was observed from an integration between the full irrigation supplement ($I_3$), vermicompost ($O_3$), and nano-Zn foliar spraying at a rate of 2 g $L^{-1}$ ($zn_3$) and the highest irrigation water productivity for straw (1.33 and 1.38 kg straw $m^{-3}$) was observed from an integration between the full irrigation supplement ($I_3$), vermicompost ($O_3$) and the absence of nano-Zn foliar spraying in the first and second seasons, respectively (Supplementary Figure S1). This could be attributed to a decrease in the water applied and improving the yield of barley production by combining the application of vermicompost and nano-Zn foliar spraying under deficit irrigation.

### 3.3. Effect of Organic Soil Amendments and Nano-Zn Foliar Spraying on Some Soil Characteristics under Irrigation Supplements

In general, applying organic amendments improved the soil's physical and chemical properties (Table 6), with the optimal improvement found for the combination of full irrigation supplement and vermicompost. Reduction in irrigation supplement ($I_2$ and $I_3$) resulted in a significant increase in (ECe, ESP, and BD), and decreased in (SOC, AN, and FC) in soil compared with full irrigation supplement $I_1$ (Table 6). Over the two growing seasons, application of $I_3$ increased soil ECe, ESP, and BD, by 20.82%, 20.89%, and 2.26%, respectively, compared with $I_1$ treatment. The SOC, AN, and FC values decreased by −16.92%, −10.65%, and −8.97%, respectively, in response to DT compared with $I_1$. The results illustrate that drought stress causes deterioration of soil properties. These findings are similar to those obtained by [63,64], who reported that the reduced water availability under low irrigation levels caused a significant decrease in field capacity, organic matter, and soil nutrient contents and significantly increased soil salinity and soil bulk density.

**Table 6.** Mean values of some physicochemical properties as affected by different organic amendments and foliar spraying applications under irrigation supplements for 2018/19 and 2019/20 seasons.

| Properties Treatment | ECe dS m$^{-1}$ | ESP | BD Mg m$^{-3}$ | SOC % | A.N mg kg$^{-1}$ | F.C. % |
|---|---|---|---|---|---|---|
| Irrigation supplements (I) | | | | | | |
| I$_1$ | 3.41 | 9.19 | 1.33 | 0.65 | 38.58 | 44.5 |
| I$_2$ | 3.87 | 10.43 | 1.34 | 0.61 | 37.06 | 43.34 |
| I$_3$ | 4.12 | 11.11 | 1.36 | 0.54 | 34.47 | 40.51 |
| F-test | ** | ** | ** | ** | ** | ** |
| LSD | 0.09 | 0.23 | 0.003 | 0.006 | 0.21 | 0.42 |
| Organic amendments (O) | | | | | | |
| O$_1$ | 3.88 | 10.48 | 1.36 | 0.48 | 32.18 | 38.55 |
| O$_2$ | 3.86 | 10.34 | 1.34 | 0.64 | 37.93 | 44.06 |
| O$_3$ | 3.66 | 9.91 | 1.33 | 0.69 | 40.00 | 45.74 |
| F-test | ** | ** | ** | ** | ** | ** |
| LSD | 0.06 | 0.14 | 0.002 | 0.005 | 0.21 | 0.42 |
| Nano-Zn foliar spraying (Zn) | | | | | | |
| Zn$_1$ | 3.84 | 10.43 | 1.35 | 0.61 | 36.35 | 42.41 |
| Zn$_2$ | 3.79 | 10.23 | 1.35 | 0.60 | 36.74 | 42.74 |
| Zn$_3$ | 3.77 | 10.07 | 1.34 | 0.59 | 37.02 | 43.21 |
| F-test | NS | NS | * | ** | ** | ** |
| LSD | NS | NS | 0.007 | 0.003 | 0.19 | 0.24 |
| The interaction (F-test) | | | | | | |
| I*O | NS | NS | ** | NS | NS | * |
| I*Zn | NS | NS | NS | NS | NS | * |
| O*Zn | -NS | NS | NS | NS | NS | NS |
| I*O* Zn | NS | NS | NS | NS | NS | NS |

Abbreviations: soil salinity (ECe), exchangeable sodium percentage (ESP), soil bulk density (BD), soil organic carbon (SOC), available nitrogen (A.N), and field capacity (F.C.). While *: refers to significant, and **: refers to highly significant differences between them ($p < 0.05, 0.01$), NS: refers to non-significant.

Vermicompost application is more effective than compost application during the growing season (Table 6). Application of vermicompost led to a significant reduction in ECe, ESP, and BD, by −5.67%, −5.44%, and −2.21%, respectively, whereas it significantly increased SOC, AN, and FC by 43.75%, 14.37%, and 18.65%, respectively, compared with unamended soil (Table 6) indicating the possibility of using VC as an appropriate organic amendment to improve physical and chemical properties of soil. Vermicompost has been optimally applied to improve the physical and chemical properties of the soil by increasing organic matter and SOC [64–69] led to reduction in EC, ESP, and BD, whereas it increased SOC, soil moisture content and nutrient contents.

Foliar spraying by nano-Zn did not have significant effects on the ECe and ESP of the soils, but foliar spraying by nano-Zn, especially 2 g L$^{-1}$, decreased the soil bulk density and significantly increased SOC, AN, and FC (Table 6). These results are in good agreement with those reported by [70]. It is noticed that foliar application of nano-ZnO has no effects on soil salinity and significantly increased available nitrogen.

The interactions between irrigation supplements, organic amendments, and nano-Zn foliar spraying did not have significant effects on soil properties (Figure 7). However, the application of vermicompost not only compensated for the estimated reduction of soil bulk density under full irrigation requirements I$_1$ (1.31 Mg m$^{-3}$) but also increased field capacity value 47.53% as compared to the initial values as shown in (Table 1).

The integration between irrigation supplements, organic amendments, and nano-Zn foliar spraying resulted in a linear relationship between BD, FC, and OM (Figure 8). Increasing OM due to organic amendment applications showed an increase in field capacity and a decrease in soil bulk density. This could confirm the importance of organic amendments as a method for environmental amelioration of saline soil.

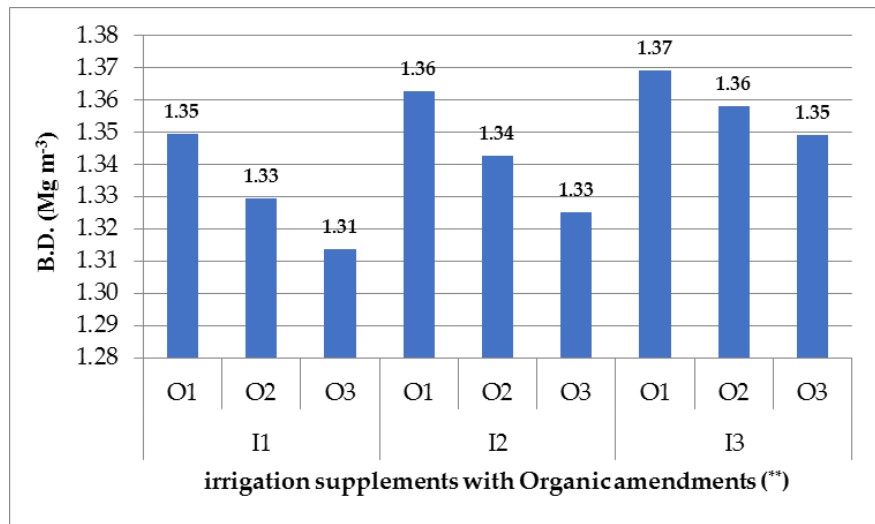

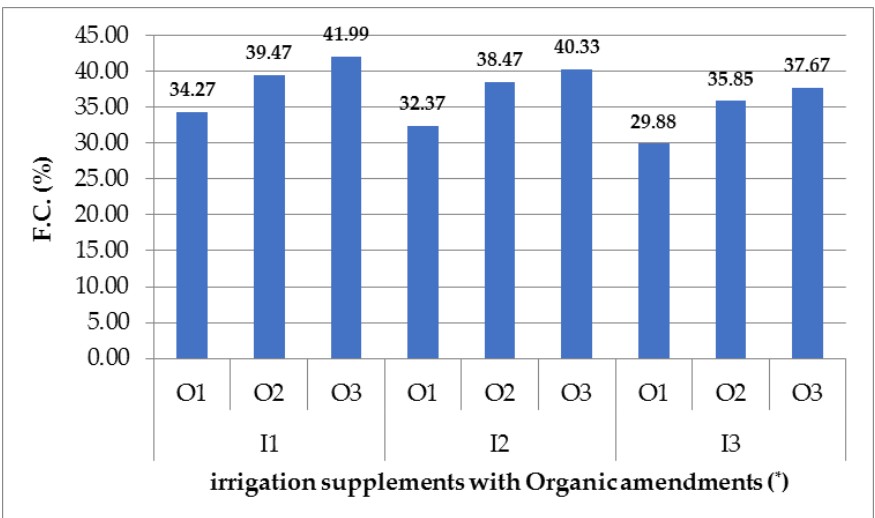

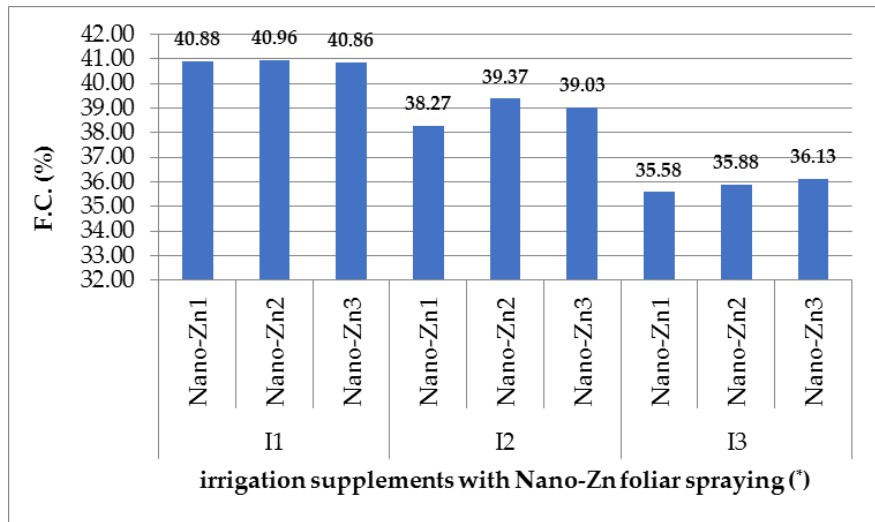

**Figure 7.** The combined effect of different treatments (irrigation supplements, organic amendments, and nano-Zn foliar spraying (average data over the two seasons) on soil bulk density (BD, Mg m$^{-1}$) and field capacity (FC, %). The (*): refers to significant, and (**): refers to highly significant differences between them ($p < 0.05$, 0.01).

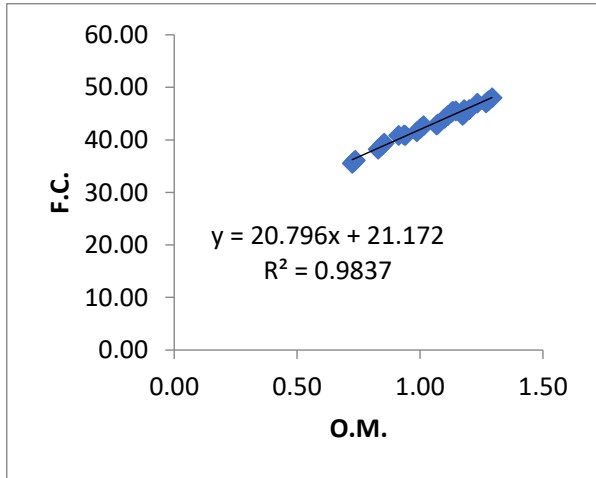
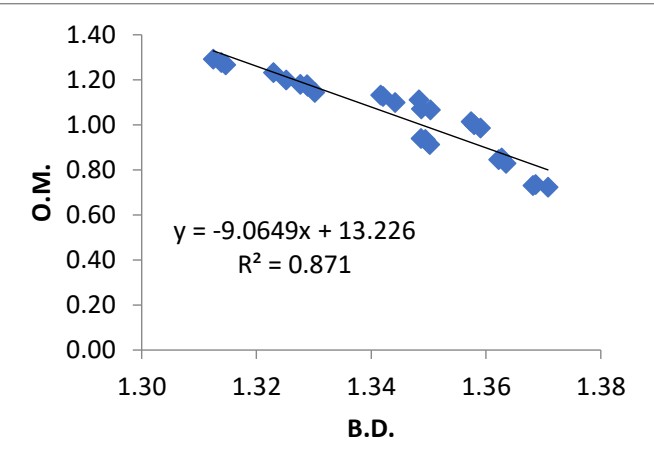

**Figure 8.** A linear relationship between soil organic matter, field capacity, and bulk density. The data represent averaged values of two seasons under all treatments.

*3.4. Relationship among All Studied Traits as Affected by the Interaction among Irrigation Supplements, Organic Amendment, and Nano-Zn Foliar Spraying Treatments*

To understand the relationships among all studied traits which affect by the interaction among irrigation supplements, organic amendment, and nano-Zn foliar spraying treatments, Pearson's correlation coefficients and cluster heat map visualization were performed and are graphically presented in Figures 9 and 10.

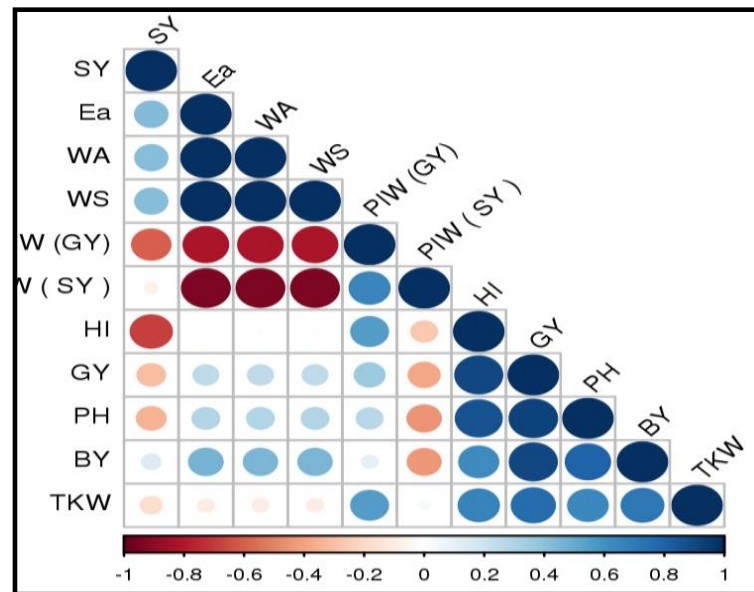

**Figure 9.** Pearson correlation coefficient heat map among grain yield, agronomical and water relations traits as affected by the interaction among irrigation supplements, organic amendment, and nano-Zn foliar spraying. Correlation key and the scale read, red circle indicated negative correlation, and blue circle indicated positive correlation, white circle mean no correlation, smaller circle indicated lesser significance and bigger circle indicated greater significance. The color intensity and size of the circle are r4lativ to the correlation coefficients. Abbreviations: Biological yield (BY), grain yield (GY), straw yield (SY), 1000-kernels weight (TKW), plant height (PH), harvest index % (HI), total water applied (W.A), stored water (W.S), water application efficiency (Ea), irrigation water productivity for grain and straw (PIWGY) (PIW SY).

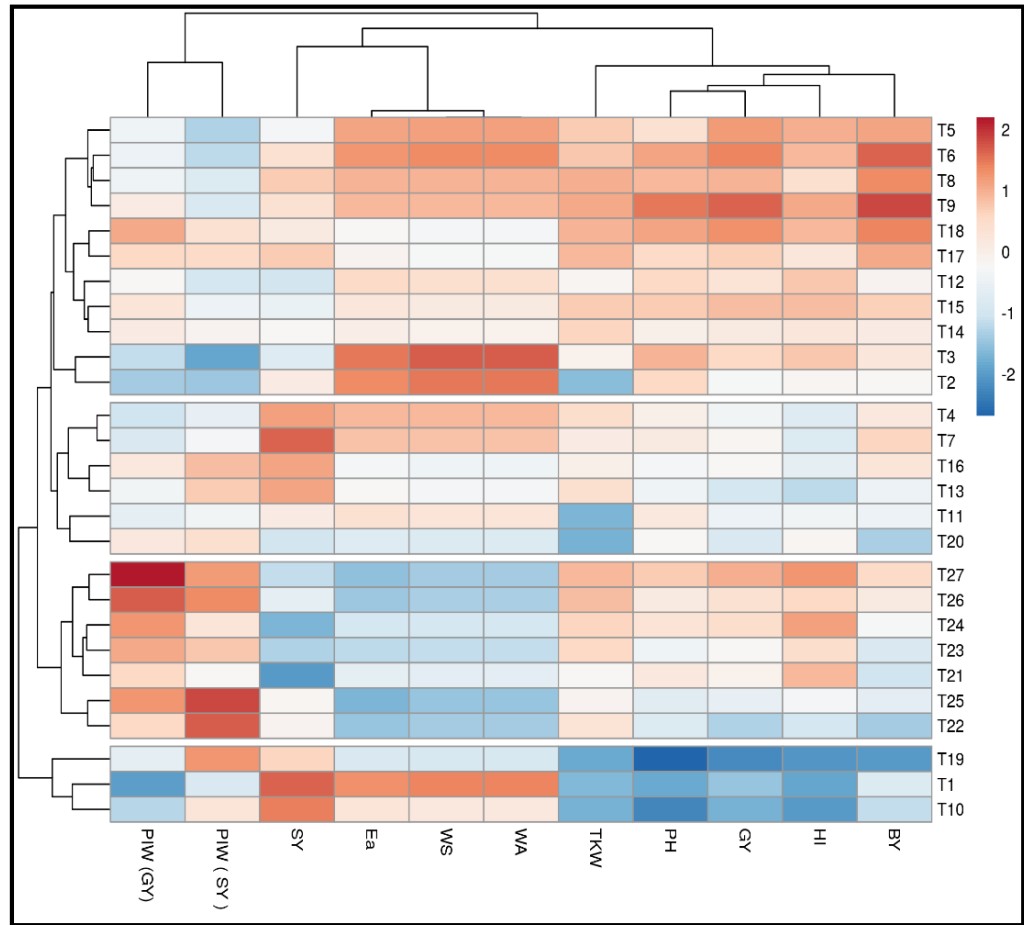

**Figure 10.** Hierarchical clustering heat map visualization of compound T27 treatments, for six agronomical and five water relation traits, showing the interaction among irrigation supplements, organic amendment, and nano-Zn foliar spraying on yield. The red color represents high values and blue color represents low values. High to low values are scaled according to the key above. Abbreviations: Biological yield (BY), grain yield (GY), straw yield (SY), 1000-kernels weight (TKW), plant height (PH), harvest index% (HI), Total water applied (W.A), stored water (W.S), water application efficiency (Ea), Irrigation water productivity for grain and straw (PIWGY) (PIW SY). $T_1$ to $T_{27}$ refer to Table 2.

The results of Pearson's correlation coefficients are presented in Figure 9. The data show that the grain yield has a high positive correlation (at $p < 0.05$, 0.01) with BY, TKW, PH, WS, WA, PIW (GY), Ea, and HI, and a negative one with SY and PIW (SY). These results indicate that improvements in grain yield under study can be created by increasing all of the WA and PIW (GY). These results were in agreement with [71,72] who reported that increasing Water Productivity (WIP) may be the best way to achieve efficient water use for barley grain yield under water stress.

Multivariate compound treatment analysis can be used to provide more information about the best treatments to help the plant breeder, as exemplified programs for water stress with detailed heat maps constructed using R software [73]. Figure 10 shows the dendrogram which was obtained by using hierarchical clustering (Euclidean distance and average linkage). The colors of the heat map represent the relationship matrix value; dark red indicates the highest values of traits, whereas the lowest values are dark blue. In the row dendrogram, all the 27 treatments were arranged in four clusters. The best treatments were $T_5$, $T_6$, $T_8$, $T_9$, $T_{18}$, $T_{17}$, $T_{12}$, $T_{15}$, $T_{14}$, $T_3$, and $T_2$; they had high mean performance values of suitable traits, revealed in the column dendrogram as an effect of the interaction among irrigation supplements, organic amendment, and nano-Zn foliar spraying treatments.



## 4. Conclusions

The results of the present study show that using organic amendments with nano-zinc foliar spraying caused significant differences in all agronomical, water relations traits and some soil properties when subjected to drought stress. Furthermore, a positive correlation was found between grain yields and water relations traits. Thus, using organic amendments with nano-zinc foliar spraying could be considered as a promising strategy for mitigating the harmful effects of drought stress in order to increase grain yield under drought conditions. The results suggest that the application of vermicompost and nano-Zn foliar spraying could be exploited to enhance barley growth and yield, using water-saving irrigation strategies, thereby alleviating some of the negative effects of drought on soil properties.

*Study Limitations*

There is not much work about the interaction between the application of organic amendments and nano-zinc, under water stress conditions, in terms of their impact on some soil properties, yield, and the water productivity of barley.

Future Directions

Future studies should focus on the effect of organic amendments with nano-zinc foliar spraying on physiological, molecular, and chemical properties of grain, and on the economics of using organic amendments with nano-zinc foliar spraying as a promising method to increasing yield under drought stress conditions to enhance farmers' income.

**Supplementary Materials:** The following supporting information can be downloaded at: https://www.mdpi.com/article/10.3390/agronomy12030585/s1, Figure S1: Total water applied, Stored water, Water application efficiency, and Irrigation water productivity for grain and straw (PIW, kg m$^{-3}$) as affected by different treatments (irrigation supplements, organic amendments, and nano-Zn foliar spraying, Table S1. The effect of the interaction between irrigation treatments and organic rates on the all studied traits in the studied seasons, Table S2: Effect of the interaction between different organic amendments and nano-Zn foliar spraying concentrations under irrigation supplements on all barley studied traits in 2018/19 and 2019/20 seasons.

**Author Contributions:** Conceptualization, T.H.K., S.A.M., Z.E.G., I.A.K. and A.A.; methodology, T.H.K., S.A.M. and Z.E.G.; software, I.A.K. and A.A.; validation, T.H.K., S.A.M., Z.E.G., I.A.K. and A.A.; formal analysis, T.H.K., S.A.M. and Z.E.G.; investigation, T.H.K., S.A.M. and Z.E.G.; resources, T.H.K., S.A.M. and Z.E.G.; data curation, T.H.K., S.A.M. and Z.E.G.; writing—original draft preparation, T.H.K., S.A.M. and Z.E.G.; writing—review and editing, I.A.K. and A.A.; visualization, T.H.K., S.A.M., Z.E.G. and A.A.; supervision, I.A.K.; funding acquisition, A.A. All authors have read and agreed to the published version of the manuscript.

**Funding:** This research received no external funding.

**Data Availability Statement:** The data and results can be available from the corresponding author upon reasonable request.

**Acknowledgments:** The authors extend their appreciation to the Soils, Water, and Environment Res. Inst., (SWERI), Barley Res., Dept. Field Crops Res. Inst., Central Lab. for Design and Stat. Anal. Res., Agriculture Research Center (ARC), Giza, Egypt, Genetics Dept., Faculty of Agriculture, Kafr El-Sheikh University, Egypt, and Dept., of Biotechnology, Faculty of Science, Taif University, Taif, Saudi Arabia. All the authors are thankful for the support provided by Labs of Soil Improvement and Conservation Res. Dep., Sakha Agri. Res. Station, Kafr El-Sheikh, Egypt.

**Conflicts of Interest:** The authors declare no conflict of interest.

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
