# Peer review of "Effect of Organic Amendments and Nano-Zinc Foliar Application on Alleviation of Water Stress in Some Soil Properties and Water Productivity of Barley Yield"

_agronomy, doi:10.3390/agronomy12030585_

Round 1

Reviewer 1 Report

Please cheque English, it seems like someone that does not speak well english made corrections previously suggested. I pointed out and corrected such mistakes up to page 16, then I stopped...

There are two many tables and figs. No need to present all statistical data. The results section must be reduced a lot.

Discussion must be added (there is almost no discussion ....).

Author Response

Point (9): Please check English, it seems like someone that does not speak well English made corrections previously suggested. I pointed out and corrected such mistakes up to page 16, then I stopped...

I corrected your suggestion, it takes the red line with yellow color

Point (10): There are two many tables and figs. No need to present all statistical data. The results section must be reduced a lot.

I reduced the results section and moved it in Supplementary Materials, and deleted the not needed statistical data

Point (11): Discussion must be added (there is almost no discussion....).

This mistake has been fixed and added discussion for all points.

Thanks for your comments and suggestions

Sincerely,

Dr. Tamer Khalifa

Reviewer 2 Report

  1. This study has investigated the application of organic amendments, and Nano-zinc, as well as their interactions on alleviation of the water stress impacts on some soil properties and the productivity of barely grain output. The background of the study is well explained. However, literature review is missing. Some studies are cited in the introduction but there is a need to highlight the important studies related to this research and briefly discuss their findings.
  2. Identify the gaps in literature. Based on the gaps authors may like to add at least one paragraph to explain the motivation of the study. Authors may also like to add expected contributions of the study, its significance, and structure of the study at the end of the introductory section.
  3. In the material and methods section, please mention the sampling technique used in the field to collect the samples.
  4. Equation numbers are missing in front of all equations. Please enter equation numbers.
  5. Section 2.5.2 through 2.5.6 do not need to number each section. All the information given in these sections can be discussed under the umbrella of water relation section 2.5.
  6. Add proper title to each table and figure in the manuscript and table notes need to be provided below each table.
  7. By listing important findings and suggesting, some recommendations based on the results may help to expand Conclusion section.
  8. If possible, study limitations and future directions should be discussed at the end of manuscript.
  9. Authors may like to check the references carefully for any incomplete information provided in reference no 14 and 23 and for any other missing information.

Author Response

Reviewer1#

Point (1): This study has investigated the application of organic amendments, and Nano-zinc, as well as their interactions on alleviation of the water stress impacts on some soil properties and the productivity of barely grain output. The background of the study is well explained. However, the literature review is missing. Some studies are cited in the introduction, but there is a need to highlight the important studies related to this research and briefly discuss their findings.

This mistake has been fixed

Point (2): Identify the gaps in the literature. Based on the gaps authors may like to add at least one paragraph to explain the motivation of the study. Authors may also like to add expected contributions of the study, its significance, and structure of the study at the end of the introductory section.

This mistake has been fixed

Point (3): In the material and methods section, please mention the sampling technique used in the field to collect the samples.

I mention the sampling technique used in the field to collect the samples in

 (2.4.3. Soil properties (page 6)

After harvest, soil samples were obtained from the surface layer (30 cm depth) using an auger and analysis of some soil properties.

Point (3): Equation numbers are missing in front of all equations. Please enter equation numbers.

This mistake has been fixed (page 6 and 7)

Point (4): Section 2.5.2 through 2.5.6 do not need to number each section. All the information given in these sections can be discussed under the umbrella of water relation section 2.5.

This mistake has been fixed (page 7)

Point (5): Add a proper title to each table and figure in the manuscript and table notes need to be provided below each table.

This mistake has been fixed

Point (6): By listing important findings and suggesting, some recommendations based on the results may help to expand the Conclusion section.

This mistake has been fixed

Point (7): If possible, study limitations and future directions should be discussed at the end of manuscript.

This mistake has been fixed, study limitations and future directions were discussed (page 19)

Point (8): Authors may like to check the references carefully for any incomplete information provided in reference no 14 and 23 and for any other missing information.

I deleted this reference from the paper

Thanks for your comments and suggestions

Sincerely,

Dr. Tamer Khalifa